# Beyond Finite Layer Neural Networks: Bridging Deep Architectures and Numerical Differential Equations

## Abstract

Deep neural networks have become the state-of-the-art models in numerous machine learning tasks. However, general guidance to network architecture design is still missing. In our work, we bridge deep neural network design with numerical differential equations. We show that many effective networks, such as ResNet, PolyNet, FractalNet and RevNet, can be interpreted as different numerical discretizations of differential equations. This finding brings us a brand new perspective on the design of effective deep architectures. We can take advantage of the rich knowledge in numerical analysis to guide us in designing new and potentially more effective deep networks. As an example, we propose a linear multi-step architecture (LM-architecture) which is inspired by the linear multi-step method solving ordinary differential equations. The LM-architecture is an effective structure that can be used on any ResNet-like networks. In particular, we demonstrate that LM-ResNet and LM-ResNeXt (i.e. the networks obtained by applying the LM-architecture on ResNet and ResNeXt respectively) can achieve noticeably higher accuracy than ResNet and ResNeXt on both CIFAR and ImageNet with comparable numbers of trainable parameters. In particular, on both CIFAR and ImageNet, LM-ResNet/LM-ResNeXt can significantly compress ($> 50\%$) the original networks while maintaining a similar performance. This can be explained mathematically using the concept of modified equation from numerical analysis. Last but not least, we also establish a connection between stochastic control and noise injection in the training process which helps to improve generalization of the networks. Furthermore, by relating stochastic training strategy with stochastic dynamic system, we can easily apply stochastic training to the networks with the LM-architecture. As an example, we introduced stochastic depth to LM-ResNet and achieve significant improvement over the original LM-ResNet on CIFAR10.

## 1 Introduction

Deep learning has achieved great success in may machine learning tasks. The end-to-end deep architectures have the ability to effectively extract features relevant to the given labels and achieve state-of-the-art accuracy in various applications (Bengio, 2009). Network design is one of the central task in deep learning. Its main objective is to grant the networks with strong generalization power using as few parameters as possible. The first ultra deep convolutional network is the ResNet (He et al., 2015b) which has skip connections to keep feature maps in different layers in the same scale and to avoid gradient vanishing. Structures other than the skip connections of the ResNet were also introduced to avoid gradient vanishing, such as the dense connections (Huang et al., 2016a), fractal path (Larsson et al., 2016) and Dirac initialization (Zagoruyko & Komodakis, 2017). Furthermore, there has been a lot of attempts to improve the accuracy of image classifications by modifying the residual blocks of the ResNet. Zagoruyko & Komodakis (2016) suggested that we need to double the number of layers of ResNet to achieve a fraction of a percent improvement of accuracy. They proposed a widened architecture that can efficiently improve the accuracy. Zhang et al. (2017) pointed out that simply modifying depth or width of ResNet might not be the best way of architecture design. Exploring structural diversity, which is an alternative dimension in network design, may lead to more effective networks. In Szegedy et al. (2017), Zhang et al. (2017), Xie et al. (2017),

Li et al. (2017) and Hu et al. (2017), the authors further improved the accuracy of the networks by carefully designing residual blocks via increasing the width of each block, changing the topology of the network and following certain empirical observations. In the literature, the network design is mainly empirical.It remains a mystery whether there is a general principle to guide the design of effective and compact deep networks.

Observe that each residual block of ResNet can be written as $u_{n+1} = u_n + \Delta t f(u_n)$ which is one step of forward Euler discretization (AppendixA.1) of the ordinary differential equation (ODE) $u_t = f(u)$ (E, 2017). This suggests that there might be a connection between discrete dynamic systems and deep networks with skip connections. In this work, we will show that many state-of-the-art deep network architectures, such as PolyNet (Zhang et al., 2017), FractalNet (Larsson et al., 2016) and RevNet (Gomez et al., 2017), can be consider as different discretizations of ODEs. From the perspective of this work, the success of these networks is mainly due to their ability to efficiently approximate dynamic systems. On a side note, differential equations is one of the most powerful tools used in low-level computer vision such as image denoising, deblurring, registration and segmentation (Osher & Paragios, 2003; Aubert & Kornprobst, 2006; Chan & Shen, 2005). This may also bring insights on the success of deep neural networks in low-level computer vision. Furthermore, the connection between architectures of deep neural networks and numerical approximations of ODEs enables us to design new and more effective deep architectures by selecting certain discrete approximations of ODEs. As an example, we design a new network structure called linear multi-step architecture (LM-architecture) which is inspired by the linear multi-step method in numerical ODEs (Ascher & Petzold, 1997). This architecture can be applied to *any ResNet-like networks*. In this paper, we apply the LM-architecture to ResNet and ResNeXt (Xie et al., 2017) and achieve noticeable improvements on CIFAR and ImageNet with comparable numbers of trainable parameters. We also explain the performance gain using the concept of modified equations from numerical analysis.

It is known in the literature that introducing randomness by injecting noise to the forward process can improve generalization of deep residual networks. This includes stochastic drop out of residual blocks (Huang et al., 2016b) and stochastic shakes of the outputs from different branches of each residual block (Gastaldi, 2017). In this work we show that any ResNet-like network with noise injection can be interpreted as a discretization of a stochastic dynamic system. This gives a relatively unified explanation to the stochastic learning process using stochastic control. Furthermore, by relating stochastic training strategy with stochastic dynamic system, we can easily apply stochastic training to the networks with the proposed LM-architecture. As an example, we introduce stochastic depth to LM-ResNet and achieve significant improvement over the original LM-ResNet on CIFAR10.

## 1.1 RELATED WORK

The link between ResNet (Figure 1(a)) and ODEs were first observed by E (2017), where the authors formulated the ODE $u_t = f(u)$ as the continuum limit of the ResNet $u_{n+1} = u_n + \Delta t f(u_n)$. Liao & Poggio (2016) bridged ResNet with recurrent neural network (RNN), where the latter is known as an approximation of dynamic systems. Sonoda & Murata (2017) and Li & Shi (2017) also regarded ResNet as dynamic systems that are the characteristic lines of a transport equation on the distribution of the data set. Similar observations were also made by Chang et al. (2017); they designed a reversible architecture to grant stability to the dynamic system. On the other hand, many deep network designs were inspired by optimization algorithms, such as the network LISTA (Gregor & LeCun, 2010) and the ADMM-Net (Yang et al., 2016). Optimization algorithms can be regarded as discretizations of various types of ODEs (Helmke & Moore, 2012), among which the simplest example is gradient flow.

Another set of important examples of dynamic systems is partial differential equations (PDEs), which have been widely used in low-level computer vision tasks such as image restoration. There were some recent attempts on combining deep learning with PDEs for various computer vision tasks, i.e. to balance handcraft modeling and data-driven modeling. Liu et al. (2010) and Liu et al. (2013) proposed to use linear combinations of a series of handcrafted PDE-terms and used optimal control methods to learn the coefficients. Later, Fang et al. (2017) extended their model to handle classification tasks and proposed an learned PDE model (L-PDE). However, for classification tasks, the dynamics (i.e. the trajectories generated by passing data to the network) should be interpreted as the characteristic lines of a PDE on the distribution of the data set. This means that using spatial

differential operators in the network is not essential for classification tasks. Furthermore, the discretizations of differential operators in the L-PDE are not trainable, which significantly reduces the network's expressive power and stability. Chen et al. (2015) proposed a feed-forward network in order to learn the optimal nonlinear anisotropic diffusion for image denoising. Unlike the previous work, their network used trainable convolution kernels instead of fixed discretizations of differential operators, and used radio basis functions to approximate the nonlinear diffusivity function. More recently, Long et al. (2017) designed a network, called PDE-Net, to learn more general evolution PDEs from sequential data. The learned PDE-Net can accurately predict the dynamical behavior of data and has the potential to reveal the underlying PDE model that drives the observed data.

In our work, we focus on a different perspective. First of all, we do not require the ODE $u_t = f(u)$ associate to any optimization problem, nor do we assume any differential structures in $f(u)$. The optimal $f(u)$ is learned for a given supervised learning task. Secondly, we draw a relatively comprehensive connection between the architectures of popular deep networks and discretization schemes of ODEs. More importantly, we demonstrate that the connection between deep networks and numerical ODEs enables us to design new and more effective deep networks. As an example, we introduce the LM-architecture to ResNet and ResNeXt which improves the accuracy of the original networks.

We also note that, our viewpoint enables us to easily explain why ResNet can achieve good accuracy by dropping out some residual blocks after training, whereas dropping off sub-sampling layers often leads to an accuracy drop (Veit et al., 2016). This is simply because each residual block is one step of the discretized ODE, and hence, dropping out some residual blocks only amounts to modifying the step size of the discrete dynamic system, while the sub-sampling layer is not a part of the ODE model. Our explanation is similar to the unrolled iterative estimation proposed by Greff et al. (2016), while the difference is that we believe it is the data-driven ODE that does the unrolled iterative estimation.

## 2 Numerical Differential Equation, Deep Networks and Beyond

In this section we show that many existing deep neural networks can be consider as different numerical schemes approximating ODEs of the form $u_t = f(u)$. Based on such observation, we introduce a new structure, called the linear multi-step architecture (LM-architecture) which is inspired by the well-known linear multi-step method in numerical ODEs. The LM-architecture can be applied to any ResNet-like networks. As an example, we apply it to ResNet and ResNeXt and demonstrate the performance gain of such modification on CIFAR and ImageNet data sets.

### 2.1 Numerical Schemes And Network Architectures

**PolyNet** (Figure 1(b)), proposed by Zhang et al. (2017), introduced a PolyInception module in each residual block to enhance the expressive power of the network. The PolyInception model includes polynomial compositions that can be described as

$$(I + F + F^2) \cdot x = x + F(x) + F(F(x)).$$

We observe that PolyInception model can be interpreted as an approximation to one step of the backward Euler (implicit) scheme (AppendixA.1):

$$u_{n+1} = (I - \Delta t f)^{-1} u_n.$$

Indeed, we can formally rewrite $(I - \Delta t f)^{-1}$ as

$$I + \Delta t f + (\Delta t f)^2 + \cdots + (\Delta t f)^n + \cdots.$$

Therefore, the architecture of PolyNet can be viewed as an approximation to the backward Euler scheme solving the ODE $u_t = f(u)$. Note that, the implicit scheme allows a larger step size (Ascher & Petzold, 1997), which in turn allows fewer numbers of residual blocks of the network. This explains why PolyNet is able to reduce depth by increasing width of each residual block to achieve state-of-the-art classification accuracy.

**FractalNet** (Larsson et al., 2016) (Figure 1(c)) is designed based on self-similarity. It is designed by repeatedly applying a simple expansion rule to generate deep networks whose structural layouts

are truncated fractals. We observe that, the macro-structure of FractalNet can be interpreted as the well-known Runge-Kutta scheme in numerical analysis. Recall that the recursive fractal structure of FractalNet can be written as $f_{c+1} = \frac{1}{2}k_c * + \frac{1}{2}f_c \circ f_c$. For simplicity of presentation, we only demonstrate the FractalNet of order 2 (i.e. $c \le 2$). Then, every block of the FractalNet (of order 2) can be expressed as

$$x_{n+1} = k_1 * x_n + k_2 * (k_3 * x_n + f_1(x_n)) + f_2(k_3 * x_n + f_1(x_n),$$

which resembles the Runge-Kutta scheme of order 2 solving the ODE $u_t = f(u)$ (see Appendix A.2).

**RevNet**(Figure 1(d)), proposed by Gomez et al. (2017), is a reversible network which does not require to store activations during forward propagations. The RevNet can be expressed as the following discrete dynamic system

$$X_{n+1} = X_n + f(Y_n),$$
$$Y_{n+1} = Y_n + g(X_{n+1}).$$

RevNet can be interpreted as a simple forward Euler approximation to the following dynamic system

$$\dot{X} = f_1(Y),$$
$$\dot{Y} = f_2(X).$$

Note that reversibility, which means we can simulate the dynamic from the end time to the initial time, is also an important notation in dynamic systems. There were also attempts to design reversible scheme in dynamic system such as Nguyen & Mcmechan (2015).

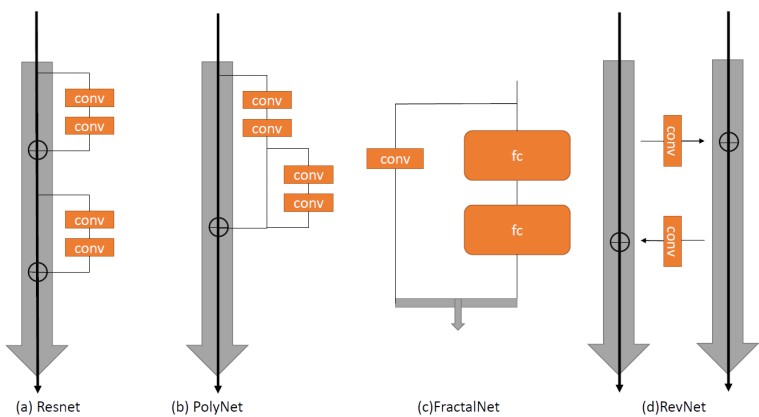

Figure 1: Schematics of network architectures.

Table 1: In this table, we list a few popular deep networks, their associated ODEs and the numerical schemes that are connected to the architecture of the networks.

| Network | Related ODE | Numerical Scheme |
|---|---|---|
| ResNet, ResNeXt, etc. | $u_t = f(u)$ | Forward Euler scheme |
| PolyNet | $u_t = f(u)$ | Approximation of backward Euler scheme |
| FractalNet | $u_t = f(u)$ | Runge-Kutta scheme |
| RevNet | $\dot{X} = f_1(Y), \dot{Y} = f_2(X)$ | Forward Euler scheme |

## 2.2 LM-RESNET: A NEW DEEP ARCHITECTURE FROM NUMERICAL DIFFERENTIAL EQUATION

We have shown that architectures of some successful deep neural networks can be interpreted as different discrete approximations of dynamic systems. In this section, we proposed a new struc-

ture, called linear multi-step structure (LM-architecture), based on the well-known linear multi-step scheme in numerical ODEs (which is briefly recalled in Appendix A.3). The LM-architecture can be written as follows

$$u_{n+1} = (1 - k_n)u_n + k_n u_{n-1} + f(u_n), \tag{1}$$

where $k_n \in \mathbb{R}$ is a trainable parameter for each layer $n$. A schematic of the LM-architecture is presented in Figure 2. Note that the midpoint and leapfrog network structures in Chang et al. (2017) are all special case of ours. The LM-architecture is a 2-step method approximating the ODE $u_t = f(u)$. Therefore, it can be applied to *any ResNet-like* networks, including those mentioned in the previous section. As an example, we apply the LM-architecture to ResNet and ResNeXt. We call these new networks the LM-ResNet and LM-ResNeXt. We trained LM-ResNet and LM-ResNeXt on CIFAR (Krizhevsky & Hinton, 2009) and Imagenet (Russakovsky et al., 2014), and both achieve improvements over the original ResNet and ResNeXt.

**Implementation Details.** For the data sets CIFAR10 and CIFAR100, we train and test our networks on the training and testing set as originally given by the data set. For ImageNet, our models are trained on the training set with 1.28 million images and evaluated on the validation set with 50k images. On CIFAR, we follow the simple data augmentation in Lee et al. (2015) for training: 4 pixels are padded on each side, and a 32×32 crop is randomly sampled from the padded image or its horizontal flip. For testing, we only evaluate the single view of the original 32×32 image. Note that the data augmentation used by ResNet (He et al., 2015b; Xie et al., 2017) is the same as Lee et al. (2015). On ImageNet, we follow the practice in Krizhevsky et al. (2012); Simonyan & Zisserman (2014). Images are resized with its shorter side randomly sampled in $[256, 480]$ for scale augmentation (Simonyan & Zisserman, 2014). The input image is $224 \times 224$ randomly cropped from a resized image using the scale and aspect ratio augmentation of Szegedy et al. (2015). For the experiments of ResNet/LM-ResNet on CIFAR, we adopt the original design of the residual block in He et al. (2016), i.e. using a small two-layer neural network as the residual block with bn-relu-conv-bn-relu-conv. The residual block of LM-ResNeXt (as well as LM-ResNet164) is the bottleneck structure used by (Xie et al., 2017) that takes the form $\begin{bmatrix} 1 \times 1, 64 \\ 3 \times 3, 64 \\ 1 \times 1, 256 \end{bmatrix}$. We start our networks with a single $3 \times 3$ conv layer, followed by 3 residual blocks, global average pooling and a fully-connected classifier. The parameters $k_n$ of the LM-architecture are initialized by random sampling from $\mathcal{U}[-0.1, 0]$. We initialize other parameters following the method introduced by He et al. (2015a). On CIFAR, we use SGD with a mini-batch size of 128, and 256 on ImageNet. During training, we apply a weight decay of 0.0001 for LM-ResNet and 0.0005 for LM-ResNeXt, and momentum of 0.9 on CIFAR. We apply a weight decay of 0.0001 and momentum of 0.9 for both LM-ResNet and LM-ResNeXt on ImageNet. For LM-ResNet on CIFAR10 (CIFAR100), we start with the learning rate of 0.1, divide it by 10 at 80 (150) and 120 (225) epochs and terminate training at 160 (300) epochs. For LM-ResNeXt on CIFAR, we start with the learning rate of 0.1 and divide it by 10 at 150 and 225 epochs, and terminate training at 300 epochs.

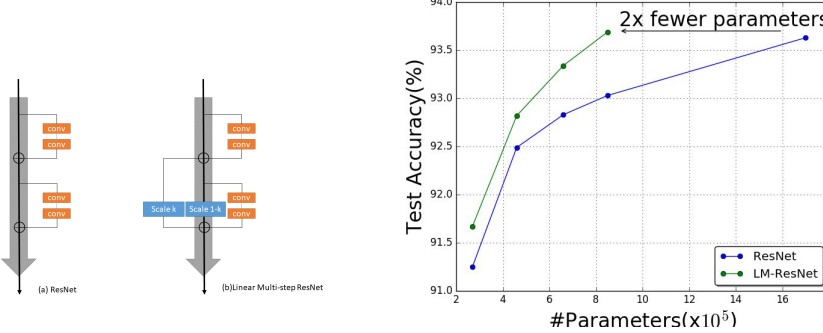

Figure 2: LM-architecture is an efficient structure that enables ResNet to achieve same level of accuracy with only half of the parameters on CIFAR10.

Table 2: Comparisons of LM-ResNet/LM-ResNeXt with other networks on CIFAR

| Model | Layer | Error | Params | Dataset |
|---|---|---|---|---|
| ResNet (He et al. (2015b)) | 20 | 8.75 | 0.27M | CIFAR10 |
| ResNet (He et al. (2015b)) | 32 | 7.51 | 0.46M | CIFAR10 |
| ResNet (He et al. (2015b)) | 44 | 7.17 | 0.66M | CIFAR10 |
| ResNet (He et al. (2015b)) | 56 | 6.97 | 0.85M | CIFAR10 |
| ResNet (He et al. (2016)) | 110, pre-act | 6.37 | 1.7M | CIFAR10 |
| LM-ResNet (Ours) | 20, pre-act | 8.33 | 0.27M | CIFAR10 |
| LM-ResNet (Ours) | 32, pre-act | 7.18 | 0.46M | CIFAR10 |
| LM-ResNet (Ours) | 44, pre-act | 6.66 | 0.66M | CIFAR10 |
| LM-ResNet (Ours) | 56, pre-act | **6.31** | 0.85M | CIFAR10 |
| ResNet (Huang et al. (2016b)) | 110, pre-act | 27.76 | 1.7M | CIFAR100 |
| ResNet (He et al. (2016)) | 164, pre-act | 24.33 | 2.55M | CIFAR100 |
| ResNet (He et al. (2016)) | 1001, pre-act | 22.71 | 18.88M | CIFAR100 |
| FractalNet (Larsson et al. (2016)) | 20 | 23.30 | 38.6M | CIFAR100 |
| FractalNet (Larsson et al. (2016)) | 40 | 22.49 | 22.9M | CIFAR100 |
| DenseNet (Huang et al., 2016a) | 100 | 19.25 | 27.2M | CIFAR100 |
| DenseNet-BC (Huang et al., 2016a) | 190 | 17.18 | 25.6M | CIFAR100 |
| ResNeXt (Xie et al. (2017)) | 29(8×64d) | 17.77 | 34.4M | CIFAR100 |
| ResNeXt (Xie et al. (2017)) | 29(16×64d) | 17.31 | 68.1M | CIFAR100 |
| ResNeXt (Our Implement) | 29(16×64d), pre-act | 17.65 | 68.1M | CIFAR100 |
| LM-ResNet (Ours) | 110, pre-act | 25.87 | 1.7M | CIFAR100 |
| LM-ResNet (Ours) | 164, pre-act | 22.90 | 2.55M | CIFAR100 |
| LM-ResNeXt (Ours) | 29(8×64d), pre-act | 17.49 | 35.1M | CIFAR100 |
| LM-ResNeXt (Ours) | 29(16×64d), pre-act | **16.79** | 68.8M | CIFAR100 |

**Results.** Testing errors of our proposed LM-ResNet/LM-ResNeXt and some other deep networks on CIFAR are presented in Table 2. Figure 2 shows the overall improvements of LM-ResNet over ResNet on CIFAR10 with varied number of layers. We also observe noticeable improvements of both LM-ResNet and LM-ResNeXt on CIFAR100. Xie et al. (2017) claimed that ResNeXt can achieve lower testing error without pre-activation (pre-act). However, our results show that LM-ResNeXt with pre-act achieves lower testing errors even than the original ResNeXt without pre-act. Training and testing curves of LM-ResNeXt are plotted in Figure3. In Table 2, we also present testing errors of FractalNet and DenseNet (Huang et al., 2016a) on CIFAR 100. We can see that our proposed LM-ResNeXt29 has the best result. Comparisons between LM-ResNet and ResNet on ImageNet are presented in Table 3. The LM-ResNet shows improvement over ResNet with comparable number of trainable parameters. Note that the results of ResNet on ImageNet are obtained from "https://github.com/KaimingHe/deep-residual-networks". It is worth noticing that the testing error of the 56-layer LM-ResNet is comparable to that of the 110-layer ResNet on CIFAR10; the testing error of the 164-layer LM-ResNet is comparable to that of the 1001-layer ResNet on CIFAR100; the testing error of the 50-layer LM-ResNet is comparable to that of the 101-layer ResNet on ImageNet. We have similar results on LM-ResNeXt and ResNeXt as well. These results indicate that the LM-architecture can greatly compress ResNet/ResNeXt without losing much of the performance. We will justify this mathematically at the end of this section using the concept of modified equations from numerical analysis.

**Explanation on the performance boost via *modified equations*.** Given a numerical scheme approximating a differential equation, its associated *modified equation* (Warming & Hyett, 1974) is another differential equation to which the numerical scheme approximates with higher order of accuracy than the original equation. Modified equations are used to describe numerical behaviors of numerical schemes. For example, consider the simple 1-dimensional transport equation $u_t = cu_x$.

Table 3: Single-crop error rate on ImageNet (validation set)

| Model | Layer | top-1 | top-5 |
|-------|-------|-------|-------|
| ResNet (He et al. (2015b)) | 50 | 24.7 | 7.8 |
| ResNet (He et al. (2015b)) | 101 | 23.6 | 7.1 |
| ResNet (He et al. (2015b)) | 152 | 23.0 | 6.7 |
| LM-ResNet (Ours) | 50, pre-act | 23.8 | 7.0 |
| LM-ResNet (Ours) | 101, pre-act | **22.6** | **6.4** |

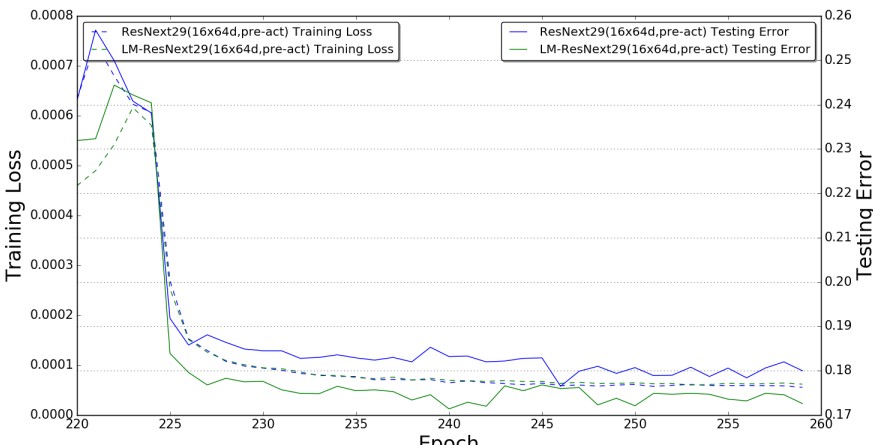

Figure 3: Training and testing curves of ResNext29 (16x64d, pre-act) and and LM-ResNet29 (16x64d, pre-act) on CIFAR100, which shows that the LM-ResNeXt can achieve higher accuracy than ResNeXt.

Both the Lax-Friedrichs scheme and Lax-Wendroff scheme approximates the transport equation. However, the associated modified equations of Lax-Friedrichs and Lax-Wendroff are

$$u_t - cu_x = \frac{\Delta x^2}{2\Delta t}(1 - r^2)u_{xx} \quad \text{and} \quad u_t - cu_x = \frac{c\Delta x^2}{6}(r^2 - 1)u_{xxx}$$

respectively, where $r = \frac{2\Delta t}{\Delta x}$. This shows that the Lax-Friedrichs scheme behaves diffusive, while the Lax-Wendroff scheme behaves dispersive. Consider the forward Euler scheme which is associated to ResNet, $\frac{u_{n+1}-u_n}{\Delta t} = f(u_n)$. Note that

$$\frac{u_{n+1} - u_n}{\Delta t} = \dot{u}_n + \frac{1}{2}\ddot{u}_n\Delta t + \frac{1}{6}\dddot{u}_n\Delta t^2 + O(\Delta t^3).$$

Thus, the modified equation of forward Euler scheme reads as

$$\dot{u}_n + \frac{\Delta t}{2}\ddot{u}_n = f(u_n). \tag{2}$$

Consider the numerical scheme used in the LM-structure $\frac{u_{n+1}-(1-k_n)u_n-k_n u_{n-1}}{\Delta t} = f(u_n)$. By Taylor's expansion, we have

$$\frac{u_{n+1} - (1 - k_n)u_n - k_n u_{n-1}}{\Delta t}$$

$$= \frac{(u_n + \Delta t\dot{u}_n + \frac{1}{2}\Delta t^2\ddot{u}_n + O(\Delta t^3))) - (1 - k_n)u_n - k_n(u_n - \Delta t\dot{u}_n + \frac{1}{2}\Delta t^2\ddot{u}_n + O(\Delta t^3)))}{\Delta t}$$

$$= (1 + k_n)\dot{u}_n + \frac{1 - k_n}{2}\Delta t\ddot{u}_n + O(\Delta t^2).$$

Then, the modified equation of the numerical scheme associated to the LM-structure

$$(1 + k_n)\dot{u}_n + (1 - k_n)\frac{\Delta t}{2}\ddot{u}_n = f(u_n). \qquad (3)$$

Comparing (2) with (3), we can see that when $k_n \leq 0$, the second order term $\ddot{u}$ of (3) is bigger than that of (2). The term $\ddot{u}$ represents acceleration which leads to acceleration of the convergence of $u_n$ when $f = -\nabla g$ Su & Boyd (2015); Wilson et al. (2016). When $f(u) = \mathcal{L}(u)$ with $\mathcal{L}$ being an elliptic operator, the term $\ddot{u}$ introduce dispersion on top of the dissipation, which speeds up the flow of $u_n$ (Dong et al., 2017). In fact, this is our original motivation of introducing the LM-architecture (1). Note that when the dynamic is truly a gradient flow, i.e. $f = -\nabla g$, the difference equation of the LM-structure has a stability condition $-1 \leq k_n \leq 1$. In our experiments, we do observe that most of the coefficients are lying in $(-1, 1)$ (Figure 4). Moreover, the network is indeed accelerating at the end of the dynamic, for the learned parameters $\{k_n\}$ are negative and close to $-1$ (Figure 4).

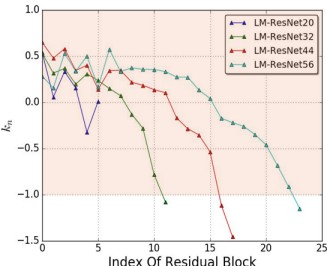 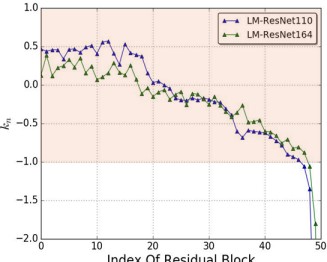

Figure 4: The trained parameters $\{k_n\}$ of LM-ResNet on CIFAR100.

## 3 STOCHASTIC LEARNING STRATEGY: A STOCHASTIC DYNAMIC SYSTEM PERSPECTIVE

Although the original ResNet (He et al., 2015b) did not use dropout, several work (Huang et al., 2016b; Gastaldi, 2017) showed that it is also beneficial to inject noise during training. In this section we show that we can regard such stochastic learning strategy as an approximation to a stochastic dynamic system. We hope the stochastic dynamic system perspective can shed lights on the discovery of a guiding principle on stochastic learning strategies. To demonstrate the advantage of bridging stochastic dynamic system with stochastic learning strategy, we introduce stochastic depth during training of LM-ResNet. Our results indicate that the networks with proposed LM-architecture can also greatly benefit from stochastic learning strategies.

### 3.1 NOISE INJECTION AND STOCHASTIC DYNAMIC SYSTEMS

As an example, we show that the two stochastic learning methods introduced in Huang et al. (2016b) and Gastaldi (2017) can be considered as weak approximations of stochastic dynamic systems.

**Shake-Shake Regularization.** Gastaldi (2017) introduced a stochastic affine combination of multiple branches in a residual block, which can be expressed as

$$X_{n+1} = X_n + \eta f_1(X_n) + (1 - \eta)f_2(X_n),$$

where $\eta \sim \mathcal{U}(0, 1)$. To find its corresponding stochastic dynamic system, we incorporate the time step size $\Delta t$ and consider

$$X_{n+1} = X_n + \left(\frac{\Delta t}{2} + \sqrt{\Delta t}(\eta - \frac{1}{2})\right)f_1(X_n) + \left(\frac{\Delta t}{2} + \sqrt{\Delta t}(\frac{1}{2} - \eta)\right)f_2(X_n), \qquad (4)$$

which reduces to the shake-shake regularization when $\Delta t = 1$. The above equation can be rewritten as

$$X_{n+1} = X_n + \frac{\Delta t}{2}(f_1(X_n) + f_2(X_n)) + \sqrt{\Delta t}(\eta - \frac{1}{2})(f_1(X_n) - f_2(X_n)).$$

Since the random variable $(\eta - \frac{1}{2}) \sim \mathcal{U}(-\frac{1}{2}, \frac{1}{2})$, following the discussion in Appendix B, the network of the shake-shake regularization is a weak approximation of the stochastic dynamic system

$$dX = \frac{1}{2}(f_1(X) + f_2(X))dt + \frac{1}{\sqrt{12}}(f_1(X) - f_2(X)) \odot [\mathbf{1}_{N \times 1}, \mathbf{0}_{N, N-1}]dB_t,$$

where $dB_t$ is an $N$ dimensional Brownian motion, $\mathbf{1}_{N \times 1}$ is an $N$-dimensional vector whose elements are all 1s, $N$ is the dimension of $X$ and $f_i(X)$, and $\odot$ denotes the pointwise product of vectors. Note from (4) that we have alternatives to the original shake-shake regularization if we choose $\Delta t \neq 1$.

**Stochastic Depth.** Huang et al. (2016b) randomly drops out residual blocks during training in order to reduce training time and improve robustness of the learned network. We can write the forward propagation as

$$X_{n+1} = X_n + \eta_n f(X_n),$$

where $\mathbb{P}(\eta_n = 1) = p_n, \mathbb{P}(\eta_n = 0) = 1 - p_n$. By incorporating $\Delta t$, we consider

$$X_{n+1} = X_n + \Delta t p_n f(X_n) + \sqrt{\Delta t} \frac{\eta_n - p_n}{\sqrt{p_n(1 - p_n)}} \sqrt{p_n(1 - p_n)} f(X_n),$$

which reduces to the original stochastic drop out training when $\Delta t = 1$. The variance of $\frac{\eta_n - p_n}{\sqrt{p_n(1-p_n)}}$ is 1. If we further assume that $(1 - 2p_n) = O(\sqrt{\Delta t})$, the condition(5) of Appendix B.2 is satisfied for small $\Delta t$. Then, following the discussion in Appendix B, the network with stochastic drop out can be seen as a weak approximation to the stochastic dynamic system

$$dX = p(t)f(X)dt + \sqrt{p(t)(1 - p(t))}f(X) \odot [\mathbf{1}_{N \times 1}, \mathbf{0}_{N, N-1}]dB_t.$$

Note that the assumption $(1 - 2p_n) = O(\sqrt{\Delta t})$ also suggests that we should set $p_n$ closer to $1/2$ for deeper blocks of the network, which coincides with the observation made by Huang et al. (2016b, Figure 8).

In general, we can interpret stochastic training procedures as approximations of the following stochastic control problem with running cost

$$\min \mathbb{E}_{X(0) \sim data} \left( \mathbb{E}(L(X(T)) + \int_0^T R(\theta)) \right)$$

$$s.t. \ dX = f(X, \theta)dt + g(X, \theta)dB_t$$

where $L(\cdot)$ is the loss function, $T$ is the terminal time of the stochastic process, and $R$ is a regularization term.

Table 4: Test on stochastic training strategy on CIFAR10

| Model | Layer | Training Strategy | Error |
|---|---|---|---|
| ResNet(He et al. (2015b)) | 110 | Original | 6.61 |
| ResNet(He et al. (2016)) | 110,pre-act | Orignial | 6.37 |
| ResNet(Huang et al. (2016b)) | 56 | Stochastic depth | 5.66 |
| ResNet(Our Implement) | 56,pre-act | Stochastic depth | 5.55 |
| ResNet(Huang et al. (2016b)) | 110 | Stochastic depth | 5.25 |
| ResNet(Huang et al. (2016b)) | 1202 | Stochastic depth | 4.91 |
| LM-ResNet(Ours) | 56,pre-act | Stochastic depth | 5.14 |
| LM-ResNet(Ours) | 110,pre-act | Stochastic depth | **4.80** |

### 3.2 STOCHASTIC TRAINING FOR NETWORKS WITH LM-ARCHITECTURE

In this section, we extend the stochastic depth training strategy to networks with the proposed LM-architecture. In order to apply the theory of Itô process, we consider the 2nd order $\ddot{X} + g(t)\dot{X} = f(X)$ (which is related to the modified equation of the LM-structure (3)) and rewrite it as a 1st order ODE system

$$\dot{X} = Y, \quad \dot{Y} = f(X) - g(t)Y.$$

Following a similar argument as in the previous section, we obtain the following stochastic process

$$\dot{X} = Y, \quad \dot{Y} = p(t)f(X)dt + \sqrt{p(t)(1-p(t))}f(X) \odot [\mathbf{1}_{N\times 1}, \mathbf{0}_{N,N-1}]dB_t - g(t)Ydt,$$

which can be weakly approximated by

$$Y_{n+1} = \frac{X_{n+1} - X_n}{\Delta t},$$

$$Y_{n+1} - Y_n = \Delta t p_n f(X_n) + \sqrt{\Delta t}(\eta_n - p_n)f(X_n) + g_n Y_n \Delta t,$$

where $\mathbb{P}(\eta_n = 1) = p_n, \mathbb{P}(\eta_n = 0) = 1 - p_n$. Taking $\Delta t = 1$, we obtain the following stochastic training strategy for LM-architecture

$$X_{n+1} = (2 + g_n)X_n - (1 + g_n)X_{n-1} + \eta_n f(X_n).$$

The above derivation suggests that the stochastic learning for networks using LM-architecture can be implemented simply by randomly dropping out the residual block with probability $p$.

**Implementation Details.** We test LM-ResNet with stochastic training strategy on CIFAR10. In our experiments, all hyper-parameters are selected exactly the same as in (Huang et al., 2016b). The probability of dropping out a residual block at each layer is a linear function of the layer, i.e. we set the probability of dropping the current residual block as $\frac{l}{L}(1 - p_L)$, where $l$ is the current layer of the network, $L$ is the depth of the network and $p_L$ is the dropping out probability of the previous layer. In our experiments, we select $p_L = 0.8$ for LM-ResNet56 and $p_L = 0.5$ for LM-ResNet110. During training with SGD, the initial learning rate is 0.1, and is divided by a factor of 10 after epoch 250 and 375, and terminated at 500 epochs. In addition, we use a weight decay of 0.0001 and a momentum of 0.9.

**Results.** Testing errors are presented in Table 4. Training and testing curves of LM-ResNet with stochastic depth are plotted in Figure5. Note that LM-ResNet110 with stochastic depth training strategy achieved a 4.80% testing error on CIFAR10, which is even lower that the ResNet1202 reported in the original paper. The benefit of stochastic training has been explained from difference perspectives, such as Bayesian (Kingma et al., 2015) and information theory (Shwartz-Ziv & Tishby, 2017; Achille & Soatto, 2016). The stochastic Brownian motion involved in the aforementioned stochastic dynamic systems introduces diffusion which leads to information gain and robustness.

## 4 CONCLUSION AND DISCUSSION

In this paper, we draw a relatively comprehensive connection between the architectures of popular deep networks and discretizations of ODEs. Such connection enables us to design new and more effective deep networks. As an example, we introduce the LM-architecture to ResNet and ResNeXt which improves the accuracy of the original networks, and the proposed networks also outperform FractalNet and DenseNet on CIFAR100. In addition, we demonstrate that networks with stochastic training process can be interpreted as a weak approximation to stochastic dynamic systems. Thus, networks with stochastic learning strategy can be casted as a stochastic control problem, which we hope to shed lights on the discovery of a guiding principle on the stochastic training process. As an example, we introduce stochastic depth to LM-ResNet and achieve significant improvement over the original LM-ResNet on CIFAR10.

As for our future work, if ODEs are considered as the continuum limits of deep neural networks (neural networks with infinite layers), more tools from mathematical analysis can be used in the study of neural networks. We can apply geometry insights, physical laws or smart design of numerical schemes to the design of more effective deep neural networks. On the other hand, numerical methods in control theory may inspire new optimization algorithms for network training. Moreover, stochastic control gives us a new perspective on the analysis of noise injections during network training.

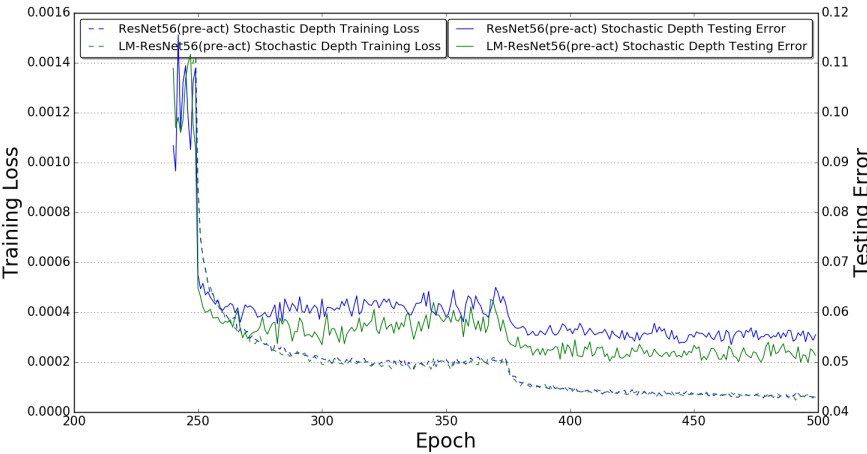

Figure 5: Training and testing curves of ResNet56 (pre-act) and and LM-ResNet56 (pre-act) on CIFAR10 with stochastic depth.

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

# APPENDIX

## A    NUMERICAL ODE

In this section we briefly recall some concepts from numerical ODEs that are used in this paper. The ODE we consider takes the form $u_t = f(u, t)$. Interested readers should consult Ascher & Petzold (1997) for a comprehensive introduction to the subject.

### A.1    FORWARD AND BACKWARD EULER METHOD

The simplest approximation of $u_t = f(u, t)$ is to discretize the time derivative $u_t$ by $\frac{u_{n+1} - u_n}{\Delta t}$ and approximate the right hand side by $f(u_n, t_n)$. This leads to the forward (explicit) Euler scheme

$$u_{n+1} = u_n + \Delta t f(u_n, t_n).$$

If we approximate the right hand side of the ODE by $f(u_{n+1}, t_{n+1})$, we obtain the backward (implicit) Euler scheme

$$u_{n+1} = u_n + \Delta t f(u_{n+1}, t_{n+1}).$$

The backward Euler scheme has better stability property than the forward Euler, though we need to solve a nonlinear equation at each step.

### A.2    RUNGE-KUTTA METHOD

Runge-Kutta method is a set of higher order one step methods, which can be formulate as

$$\hat{u}_i = u_n + \Delta t \sum_{j=1}^{s} a_{ij} f(\hat{u}_j, t_n + c_j \Delta t),$$

$$u_{n+1} = u_n + \Delta t \sum_{j=1}^{s} b_j f(\hat{u}_j, t_n + c_j \Delta t).$$

Here, $\hat{u}_j$ is an intermediate approximation to the solution at time $t_n + c_j \Delta t$, and the coefficients $\{c_j\}$ can be adjusted to achieve higher order accuracy. As an example, the popular 2nd-order Runge-Kutta takes the form

$$\hat{x}_{n+1} = x_n + \Delta t f(x_n, t_n),$$

$$x_{n+1} = x_n + \frac{\Delta t}{2} f(x_n, t_n) + \frac{\Delta t}{2} f(\hat{x}_{n+1}, t_{n+1}).$$

### A.3    LINEAR MULTI-STEP METHOD

Liear multi-step method generalizes the classical forward Euler scheme to higher orders. The general form of a $k-$step linear multi-step method is given by

$$\sum_{j=0}^{k} \alpha_j u_{n-j} = \Delta t \sum_{j=0}^{k-1} \beta_j f_{u_{n-j}, t_{n-j}},$$

where, $\alpha_j, \beta_j$ are scalar parameters and $\alpha_0 \neq 0, |\alpha_j| + |\beta_j| \neq 0$. The linear multi-step method is explicit if $\beta_0 = 0$, which is what we used to design the linear multi-step structure.

## B    NUMERICAL SCHEMES FOR STOCHASTIC DIFFERENTIAL EQUATIONS

### B.1    ITÔ PROCESS

In this section we follow the setting of Kesendal (2000) and Evans (2013). We first give the definition of Brownian motion. The Brownian motion $B_t$ is a stochastic process satisfies the following assumptions

- $B_0 = 0$ a.s.,
- $B_t - B_s$ is $\mathcal{N}(0, t - s)$ for all $t \geq s \geq 0$,
- For all time instances $0 < t_1 < t_2 < \cdots < t_n$, the random variables $W(t_1), W(t_2) - W(t_1), \cdots, W(t_n) - W(t_{n-1})$ are independent. (Also known as independent increments.)

The Itô process $X_t$ satisfies $dX_t = f(X_t, t)dt + g(X_t, t)dB_t$, where $B_t$ denotes the standard Brownian motion. We can write $X_t$ as the following integral equation

$$X_t = X_0 + \int_0^t f(s, t, \omega)ds + \int_0^t g(s, t, \omega)dB_s$$

Here $\int_0^t g(s, w)dB_s$ is the Itô's integral which can be defined as the limit of the sum

$$\sum_i g(s(a_i), a_i, \omega)[B_{a_i} - B_{a_{i-1}}](\omega)$$

on the partition $[a_i, a_{i+1}]$ of $[0, t]$.

## B.2 WEAK CONVERGENCE OF NUMERICAL SCHEMES

Given the Itô process $X_t$ satisfying $dX_t = f(X_t, t)dt + g(X_t, t)dB_t$, where $B_t$ is the standard Brownian motion, we can approximate the equation using the forward Euler scheme (Kloeden & Pearson, 1992)

$$X_{n+1} = X_n + f(X_n, t_n)\Delta t + g(X_n, t_n)\Delta W_n.$$

Here, following the definition of Itô integral, $\Delta W_n = B_{t_{n+1}} - B_{t_n}$ is a Gaussian random variable with variance $\Delta t$. It is known that the forward Euler scheme converges strongly to the Itô process. Note from (Kloeden & Pearson, 1992, Chapter 11) that if we replace $\Delta W_n$ by a random variable $\Delta \hat{W}_n$ from a non-Gaussian distribution, the forward Euler scheme becomes the so-called *simplified weak Euler scheme*. The simplified weak Euler scheme converges weakly to the Itô process if $\Delta \hat{W}_n$ satisfies the following condition

$$|\mathbb{E}(\Delta \hat{W}_n)| + |\mathbb{E}((\Delta \hat{W}_n)^3)| + |\mathbb{E}((\Delta \hat{W}_n)^2) - \Delta t| \leq K\Delta t^2 \tag{5}$$

One can verify that the random variable from the uniform distribution over a interval with 0 mean, or a (properly scaled) Bernoulli random variable taking value 1 or -1 satisfies the condition (5).

