# OpenReview forum: "Beyond Finite Layer Neural Networks: Bridging Deep Architectures and Numerical Differential Equations"
_ICLR.cc/2018/Conference — Invite to Workshop Track_

### Official Review · AnonReviewer2 · 2017-11-21
**[Beyond Finite Layer Neural Networks: Bridging Deep Architectures and Numerical Differential Equations]**

**Rating:** 6
**Confidence:** 1

**Review:**

The authors proposed to bridge deep neural network design with numerical differential equations. They found that many effective networks can be interpreted as different numerical discretization of differential equations and provided a new perspective on the design of effective deep architectures.

This paper is interesting in general and it will be useful to design new and potentially more effective deep networks. Regarding the technical details, the reviewer has the following comments:

- The authors draw a relatively comprehensive connection between the architecture of popular deep networks and discretization schemes of ODEs. Is it possible to show stability of the architecture of deep networks based on their associated ODEs? Related to this, can we choose step size or the number of layers to guarantee numerical stability?

- It is very interesting to consider networks as stochastic dynamic systems. Are there any limitations of this interpretation or discrepancy due to the weak approximation?

---

> ### Author Response · Authors · 2017-12-04
> **Response to reviewer's original comments**
>
> First of all, we appreciate the reviewer's effort in evaluating the manuscript and his/her comments. Our responses to the reviewer's comments are as follows. We have made minor modifications to the manuscript to improve clarity (especially Section 3).
>
> Reviewer: Is it possible to show stability of the architecture of deep networks based on their associated ODEs?
>
> Our Responses: One of the cited references discussed stability [Bo Chang, Lili Meng, Eldad Haber, Lars Ruthotto, David Begert, and Elliot Holtham. Reversible architectures for arbitrarily deep residual neural networks. arXiv preprint arXiv:1709.03698, 2017]. The authors disceretized a stable ODE to construct a neural network. However, BN or other stochastic training may break the stability. In our opinion, the relavance of stability (of numerical schemes or the ODE itself) to deep architecture design is still a debatable topic. It definitely deserves further invesigation, which will be our future work.
>
> Reviewer: It is very interesting to consider networks as stochastic dynamic systems. Are there any limitations of this interpretation or discrepancy due to the weak approximation?
>
> Our Responses: It is a very interesting question. In the revised version of the manuscript, we included some discussions related to this question. In short, under suitable conditions on the parameters, we will have a weak limit. However, at this point, we are not sure whether all the conditions are indeed satisfied in practice. Nonetheless, the weak limit may shed light on the choice of the hyper-parameters, such as the drop probabilities of the stochastic depth ResNet (see our discussions on stochastic depth in Section 3.1).

---

### Official Review · AnonReviewer3 · 2017-11-24
**further comparisons & inconsistent reported baselines**

**Rating:** 5
**Confidence:** 3

**Review:**

The authors cast some of the most recent CNN designs as approximate solutions to discretized ODEs. On that basis, they propose a new type of block architecture which they evaluate on CIFAR and ImageNet. They show small gains when applying their design on the ResNet architectures. They also draw a comparison between a stochastic learning process and approximation to stochastic dynamic systems.

Pros:
(+) The paper presents a way to connect NN design with principled approximations to systems
(+) Experiments are shown on compelling benchmarks such as ImageNet
Cons:
(-) It is not clear why the proposed approach is superior to the other designs
(-) Gains are relatively small and at a price of a more complicated design
(-) Incosistent baselines reported

While the effort of presenting recent CNN designs as plausible approximations to ODEs, the paper does not try to draw connections among the different approaches, compare them or prove the limits of their related approximations. In addition, it is unclear from the paper how the proposed approach (LM-architecture) compares to the recent works, what are the benefits and gains from casting is as a direct relative to the multi-step scheme in numerical ODEs. How do the different approximations relate in terms of convergence rates, error bounds etc.?

Experimentwise, the authors show some gains on CIFAR 10/100, or 0.5% (see ResNeXt Table1), while also introducing slightly more parameters. On ImageNet1k, comparisons to ResNeXt are missing from Table3, while the comparison with the ResNets show gains in the order of 1% for top-1 accuracy.

Table3 is concerning. With a single crop testing scheme, ResNet101 is yielding top-1 error of 22% and top-5 error of 6% (see Table 5 of Xie et al, 2017 (aka ResNeXt)). However, the authors report 23.6% and 7.1% respectively for their ResNet101. The performance stated by the authors of ResNe(X)t weakens the empirical results of LM-architecture.

---

> ### Author Response · Authors · 2017-12-04
> **Response to reviewer's original comments**
>
> First of all, we appreciate the reviewer's effort in evaluating the manuscript and his/her comments. Our responses to the reviewer's comments are as follows. We have made minor modifications to the manuscript to improve clarity (especially Section 3).
>
> Reviewer: It is not clear why the proposed approach is superior to the other designs
>
> Our Responses: We have already discuss this in the paper. Most of recent work like SENet, PolyNet, Inception-v4, are focusing on the improvment of the residual block, which means improving the right-hand-side of the dynamics f(u). We explore the other
> dimension, i.e. the ways to design shortcuts by bridging some of the existing designs of shortcuts with various temporal discretization of dynamic systems. Then we introduced a new micro-structure, called the LM-structure, which can be combined with exsiting designs of f(u). For example, you could have LM-PolyNet, LM-Inception, etc. We take ResNet and ResNeXt as examples to show that the LM-structure can indeed improve accuracy and compress parameters over the original deep networks. We also showed that the LM-structure can also be combined with stochastic training strategy in Section 3.
>
> Reviewer: Gains are relatively small and at a price of a more complicated design
>
> Our Responses: The design is SIMPLE! You only need to include one more shortcut to each block of ResNet (or other similar networks). We will comment on "gains are relatively small" in our later responses.
>
> Reviewer: Incosistent baselines reported
>
> Our Responses: This is because lots of the setting is not the same, like the data augmentation setting is not the same as the [Table5 Xie et al, 2017 (aka ResNeXt)] on Imagenet. We don't apply color jitter, lighting and color normalization. Thus we don't think using the baseline in the ResNeXt paper is a fair comparsion. Moreover, the ResNeXt paper did not use the pre-act ResNet. Nonetheless, we will do more experiment to make the experiment stronger.
>
> Reviewer: Experimentwise, the authors show some gains on CIFAR 10/100, or 0.5% (see ResNeXt Table1), while
> also introducing slightly more parameters.
>
> Our Responses: For ResNeXt is not a deep network, the acceleration of the LM-structure will not bring as much benefits as deeper networks.  If we use the LM-structure in deep neural networks, the accuracy will boost significantly. LM-Resnet56 achieves comparable accuracy as LM-Resnet 110 (see ResNet Table 1) on CIFAR10. And LM-ResNet164 gains 1.4% accuracy over ResNet164 on CIFAR100, which is only 0.2% lower than the accuracy of ResNet1001.  On the other hand, the reviewer commented that our LM-ResNeXt "only" gains 0.5% accuracy boost by adding 0.7M parameters. Let's take a look at the result of ResNeXt again, ResNeXt29(16x64d) adding 34M paramters to ResNeXt29(8x64d), only gains 0.46%. Therefore, we do not think 0.5% is a small improvement for ResNeXt.

---

### Official Review · AnonReviewer1 · 2017-11-27
**Useful experiment results and not so clear insights**

**Rating:** 5
**Confidence:** 1

**Review:**

Originality
--------------
The paper takes forward the idea of correspondence between  ResNets and discretization of ODEs. Introducing multi-step discretization is novel.

Clarity
---------
1)  The paper does not define the meaning of u_n=f(u).
2) The stochastic control problem (what is the role of controller, how is connected to the training procedure) is not defined

Quality
---------
While the experiments are done in CIFAR-10 and 100,  ImageNet and improvements are reported, however, connection/insights to why the improvement is obtained is still missing. Thus the evidence is only partial, i.e., we still don't know why the connection between ODE and ResNet is helpful at all.

Significance
-----------------
Strength: LM architectures reduce the layers in some cases and achieve the same level of accuracy.
Weakness: Agreed that LM methods are better approximations of the ODEs. Where do we gain? (a) It helps if we faithfully discretize the ODE. Why does (a) help? We don't have a clear answer; which takes back to the lack of what the underlying stochastic control problem is.

---

> ### Author Response · Authors · 2017-12-04
> **Response to reviewer's original comments**
>
> First of all, we appreciate the reviewer's effort in evaluating the manuscript and his/her comments. Our responses to the reviewer's comments are as follows. We have made minor modifications to the manuscript to improve clarity (especially Section 3).
>
> Reviewer:  The paper does not define the meaning of u_n=f(u)
>
> Our Response: There should be a subscript "n" of the variable "u". We apologize for the confusion. It is a standard notation for a discrete dynamic system.
>
> Reviewer: The stochastic control problem (what is the role of controller, how is connected to the training
> procedure) is not defined
>
> Our Response: The reviewer may have missed our definition of stochastic control, which is given at the end of Section 3.1. The optimization problem is a stochastic control if we consider, for example, the ResNet with stochastic training as a stochastic dynamic system (stochastic ResNet for short). The "control" is the stochastic differential equation weakly approximated by the stochastic ResNet. For there is an expectation in the objective function, we only need a week approximation, which means we only need to approximate the distribution of data instead of each individual data (or trajectory). Stochastic control is a classical and yet important topic in applied mathematics, which has wide applications in various areas especially finance. We recommend a popular tutorial given by L. Evans "Evans L C. OPTIMAL CONTROL THEORY. Springer, 1974" for further reference.
>
> Reviewer: While the experiments are done in CIFAR-10 and 100, ImageNet and improvements are reported,
> however, connection/insights to why the improvement is obtained is still missing. Thus the evidence is
> only partial, i.e., we still don't know why the connection between ODE and ResNet is helpful at all.
>
> Our Response: The reviewer may have missed our explanation on the performance boost of the proposed LM-structure. We explained the performance boost using the concept of "modified equations" from the bottom of page 6 to page 8. Basically, we argued, both analytically and experimentally, that the proposed LM-structure can be viewed as adding a momentum to the information propagation.
>
> Reviewer: Agreed that LM methods are better approximations of the ODEs. Where do we gain? (a) It helps if we faithfully discretize the ODE. Why does (a) help? We don't have a clear answer; which takes back to the lack of what the underlying stochastic control problem is.
>
> Our Responses: As we stated a couple times in the manuscript, our purpose is not to show that we should seriously approximate dynamic systems. Our main objective is to point out that effective architectures (or topology of the networks) are similar (or identical for some cases) to discretizations of dynamic systems. This is not limited to ResNet. It is more general than what have been discovered in the past. More importantly, we are able to introduce the LM-structure, which is new and can be applied to any ResNet-like networks to significantly compress the number of parameters. If your application does not care about parameter compression, LM-structure can still further improve classification accuracy of heavy-duty networks. Finally, if stochastic training is applied, we identify it with the stochastic control problem, and it is still beneficial to apply the LM-structure to discretize the underlying stochastic differential equation. The performance boost shown by our experiments can be explained using modified equations, which to our best knowledge, is a new perspective to qualitatively evaluate deep networks that can be viewed as approximations of dynamic systems.

---

### Official Review · AnonReviewer4 · 2018-01-12
**Good intuition and derivation. Consistent increase in performance.**

**Rating:** 7
**Confidence:** 4

**Review:**

Summary
- This paper draws analogy from numerical differential equation solvers and popular residual network-like deep learning architectures. It makes connection from ResNet, FractalNet, DenseNet, and RevNet to different numerical solvers such as forward and backward Euler and Runge-Kunta. In addition, inspired by the Linear Multi-step methods (LM), the authors propose a novel LM-ResNet architecture in which the next residual block takes a linear combination of the previous two residual blocks’ activations. They also propose a stochastic version of LM-ResNet that resembles Shake-shake regularization and stochastic depth. In both deterministic and stochastic cases, they show a positive improvement in classification accuracy on standard object classification benchmarks such as CIFAR-10/100 and ImageNet.

Pros
- The intuition is good that connects differential equation and ResNet-like architecture, also explored in some of the related work.
- Building upon the intuition, the author proposes a novel architecture based on a numerical ODE solver method.
- Consistent improvement in accuracy is observed in both deterministic and stochastic cases.

Cons
- The title is a little bit misleading. “Beyond Finite Layer Neural Networks” sounds like the paper proposes some infinite layer neural networks but the paper only studies finite number of layers.
- One thing that needs to be clarified is that, if the network is not targeted at solving certain ODEs, then why is the intuition from ODE matters? The paper does not motivate readers in this perspective.
- Given the widespread use of ResNet in the vision community, the incremental improvement of 1% on ImageNet is less likely to push vision research to switch to a completely different architecture. Therefore, the potential impact of the this paper to vision community is probably limited.

Conclusion
- Based on the comments above, I think the paper is a good contribution which links ODE with Deep Networks and derivation is convincing. The proposed new architecture can be considered in future architecture designs. Although the increase in performance is small, I think it is good enough to accept.

---

> ### Author Response · Authors · 2018-01-22
> **Responses to reviewer's comments**
>
> Thanks for taking your time reviewing our manuscript and all your comments. Here are our responses to the "cons".
>
> The reason behind making the current title is to indicate the potential benefit of thinking beyond finite layer neural networks by looking at the continuum, i.e. the underlying dynamic system. This helps to guide us in the design of new architectures such as the proposed LM-architecture. In other words, we would like to convey the idea that there is benefit in "thinking analog, acting digital".
>
> Similar as above, the intuition of bridging discretization of ODEs with the shortcut designs of deep networks opens the possibility of designing new and more effective deep networks. We used LM-architecture as one example. This also enabled us to use concepts in numerical analysis to analyze some of the behavior of the networks. In our paper, we used the concept of modified equations to explain why the LM-architecture can significantly reduce network depth without dropping performance (at the end of Section 2, page 7-8) .
>
> We do agree that, with same network complexity, we only marginally gain on accuracy. However, for the moment, the major advantage of the LM architecture is the compression of deep networks (and we used modified equation to explain why we have such compression), which is important in applications where heavy-duty networks cannot be used (such as on portable devices).

---

### Decision · Program_Chairs · 2018-01-29
**ICLR 2018 Conference Acceptance Decision**

**Decision:**

Invite to Workshop Track

**Comment:**

The reviewers agree that the proposed architecture is novel. However, there are issues in terms of the motivation. It would be helpful in future drafts to strengthen the argument about why the architecture is expected to be better than others. Most importantly, the gains at this stage are still incremental. A larger improvement from the new architecture would motivate more researchers to focus on this architecture.